# Explaining Human Comparisons using Alignment-Importance Heatmaps

**Nhut Truong, Dario Pesenti & Uri Hasson**
Center for Mind/Brain Sciences (CIMeC)
University of Trento
Rovereto, Trento 38068, Italy
`{leminhnhut.truong, dario.pesenti, uri.hasson}`@unitn.it

## Abstract

We present a computational explainability approach for human comparison tasks, using Alignment Importance Score (AIS) heatmaps derived from deep-vision models. The AIS reflects a feature-map's unique contribution to the alignment between Deep Neural Network's (DNN) representational geometry and that of humans. We first validate the AIS by showing that prediction of out-of-sample human similarity judgments is improved when constructing representations using only higher-AIS feature maps identified from a training set. We then compute image-specific heatmaps that visually indicate the areas that correspond to feature-maps with higher AIS scores. These maps provide an intuitive explanation of which image areas are more important when it is compared to other images in a cohort. We observe a strong correspondence between these heatmaps and saliency maps produced by a gaze-prediction model. However, in some cases, meaningful differences emerge, as the dimensions relevant for comparison are not necessarily the most visually salient. To conclude, Alignment Importance improves prediction of human similarity judgments from DNN embeddings, and provides interpretable insights into the relevant information in image space.

## 1 Introduction

### 1.1 The question: Explaining human comparisons

Work in recent years has shown that DNNs learn feature spaces whose geometry has some similarity to that of humans. This is convincingly shown by the fact that human similarity judgments (HSJs) for pairs of words or images are often quite well predicted by the distances between image-pairs or word-pairs in vision-DNNs or language models (Battleday et al., 2021; Roads & Love, 2024; Sucholutsky et al., 2023). These models therefore naturally extract features relevant for modeling HSJs when trained on standard tasks such as image classification or word prediction. While the object-embeddings of such pretrained machine learning models approximate HSJs quite well, it has been further shown that these predictions can be considerably improved using down-stream operations.

One such operation is to learn a reweighting of the products of feature values, which improves prediction of HSJs for both images (e.g., Peterson et al., 2018; Kaniuth & Hebart, 2022) and words (e.g., Richie & Bhatia, 2021). Another approach is to use supervised pruning to assess features' importance in the context of estimating a set of similarity judgments (Tarigopula et al., 2023; Flechas Manrique et al., 2023). Pruning does not alter the activation weights of the retained features, but instead removes a subset of features from the embedding matrix. Pruning has also been used to identify sub-spaces in language models that optimize particular classification tasks (e.g., Cao et al., 2021).

While prior work has shown that pruning of nodes in a DNN's penultimate layer can improve prediction of similarity judgments, here we are interested in its potential to explain what parts of an image matter for the judgment itself. Understanding which information is used as a basis for comparison is a fundamental question in cognitive science. Since the work of Tversky (1977), many studies have shown that comparisons between objects are a function of those elements that are shared or

distinct between them. However, for naturalistic stimuli, it is difficult to know which properties are important when an image is compared to a target set of images. Here we suggest that this question is tractable via a computational solution in which latent dimensions that are related to the comparison process are identified and projected onto the image space as a heatmap. We release our code at *https://github.com/tlmnhut/ais_heatmap*

## 1.2 LOGIC OF THE CURRENT STUDY

We present the logic here, with a complete formal presentation provided in Section 2.1. Our approach relies on evaluating the alignment between human and computer-model representational spaces under controlled perturbations implemented via masking. Both spaces are operationalized using pairwise distance values between images. For a given target image $t$, one set of distances is derived from human behavior ($HB_{dist}$), while the other distance-set is computed from a computer model ($Model_{dist}$). We define the baseline isomorphism between the two spaces as the correlation between these two vectors.

In the next step, a perturbation is introduced to image $t$. Specifically, a patch of image $t$ is masked so that information from that patch is not encoded in the model. Subsequently, $Model_{dist}$ is re-computed, as is the isomorphism between the representations. Note that only the target image is masked, and not the other images. Furthermore, $HB_{dist}$ remains unchanged. The crucial aspect is understanding how masking affects the isomorphism between $Model_{dist}$ and $HB_{dist}$.

1) If the mask is applied to a part of image $t$ that is not represented in the deep layers, this will not change the isomorphism as the embeddings remain unaffected.

2) If the mask covers a portion of image $t$ whose information is reflected in the deep layers, there are two possible outcomes: i) if the encoded information from the patch is cognitively irrelevant, its removal could alter $Model_{dist}$ in a way that improves the isomorphism with human similarity judgments. Conversely, ii) if the encoded information from the patch is cognitively-relevant (e.g., masking an animal's face in context of similarity judgments between animals), its removal will alter $Model_{dist}$ in a way that decreases the isomorphism with human judgments. This occurs because the way that images stand in relation to each other in the DNN representation is now lacking information that underlies human judgments.

By sweeping a mask over the image, each masked image location is associated with a perturbation score indicating the importance of the masked patch. A study by Tarigopula et al. (2023) used this approach with human neuroimaging data to explain which parts of an image contain information relevant to the representational space of various brain regions. In other work, Palazzo et al. (2020) used a Siamese network to learn a joint embedding between vision-dnn embeddings and EEG-data embeddings collected while participants viewed the same images. This produced a compatibility score between the EEG and vision-DNN domains. The authors then masked image patches to evaluate how masking impacted the compatibility score.

## 1.3 CURRENT AIMS AND CONTRIBUTION

The current study's aims advances over prior studies in three respects: it directly studies human comparison processes, it introduces an advantageous masking procedure, and it evaluates the results against typical saliency maps. The aforementioned studies operationalized representational spaces from multivariate fMRI and EEG recordings but have not studied human comparison processes. Furthermore, the technique they use, namely, mask-sweep over an image, presents several major limitations: 1) the mask size is arbitrary, requiring the use of multiple sizes; 2) an arbitrary decision is required regarding how to combine information from different mask sizes; 3) the process is computationally costly, as masks are ideally applied with each pixel being in the mask center; 4) a theoretical weakness is that the mask is not informed by prior information contained in the model.

Departing from these prior studies, here we directly model human comparison judgments, and use a different, more efficient approach to masking images, which uses information already present in the DNNs own feature space. Specifically, we focus on the feature maps in a deep convolutional layer, and use them to define the masks. Our approach is inspired by Score-CAM (Score-weighted Class Activation Maps; Wang et al., 2020) which is an explanatory method that generates heatmaps indicating which sections of a target image are relevant for its classification. Score-CAM takes the

information in each feature map, upscales it to the original input resolution, uses it as an information selector for the original input image, and computes the activation for correct class (pre-softmax confidence) when using that feature map alone. After repeating this process for all feature maps, the confidence scores are used as weights to generate a heatmap highlighting image areas important for classification. Using a similar logic, we show that information at the feature-map level is also highly useful for identifying which feature maps are important for the alignment between the DNN and human representational spaces, and that these can be visualized in a similar manner.

Beyond our main explainability objective, we have two other important aims. First, we evaluate whether it is possible to identify feature maps that are particularly important for predicting human representational spaces; using only these feature maps should improve out of sample prediction accuracy for human similarity judgments as compared to using all feature maps. Second, we evaluate the relationship between heatmaps produced using this method, and traditional saliency maps. While the latter operationalize saliency using information latent in the image itself, the heatmaps we produce highlight information pertinent to image comparisons within a given set.

## 2 METHODS

### 2.1 PRELIMINARIES

• Architecture and datasets: We use VGG-16, a deep neural network (Simonyan & Zisserman, 2014), pre-trained on ImageNet (Deng et al., 2009) and another trained on Ecoset[1] (Mehrer et al., 2021). VGG-16 was used because Ecoset was trained on that model. As images we used a dataset provided by Peterson et al. (2018), which consists of 720 images divided into six categories of 120 images. The categories were: Animals, Fruits, Furniture, Various, Vegetables and Automobiles (the latter effectively including any means of transportation including horses, sleds, cranes; Transportation henceforth). Images had a native resolution of $500 \times 500$ which was downscaled to $224 \times 224$ to fit the model.

• Human Similarity Judgments: Let $\boldsymbol{H}$ be a matrix representing the similarity judgments provided by human assessors for $n$ objects. Each entry $H_{i,j}$ in the matrix corresponds to the similarity judgment between objects $i$ and $j$. We use the upper triangle of matrix $\boldsymbol{H}$, denoted as $\boldsymbol{H}_u$.

• Object distances in feature space: Let $\boldsymbol{C}$ be a matrix representing the embeddings of $n$ images onto $d$ features of the penultimate layer of the pre-trained computer vision model, denoted as $\boldsymbol{C} \in \mathbb{R}^{n \times d}$. Specifically, we use VGG-16 with $d = 4096$, and the number of images in each Peterson's category is $n = 120$. Matrix $\boldsymbol{C}$ is obtained by considering all parameters of the pre-trained model, and specifically all 512 feature maps of the deepest convolutional layer. $\boldsymbol{Z}_u$ is the upper triangle of image-pair similarity matrix $\boldsymbol{Z}$, computed from the Pearson correlation for each row pair in $\boldsymbol{C}$.

• Subspaces in matrix $\boldsymbol{C}$: We produce two variants of $\boldsymbol{C}$ (all with dimension $n \times d$). The first variant ("remove 1"), denoted as $\boldsymbol{C}^{(\neg k)}$, is constructed by excluding feature map $k$ where $k \in \{1, 2, \ldots, 512\}$. The second variant is produced when using only a subset $S$ of feature maps in the model. Let $S \subseteq \{1, 2, \ldots, 512\}$ be a set of selected feature-map indices, and let $\boldsymbol{C}^{(S)}$ be the matrix representing the embedding of $n$ images onto $d$ nodes in the penultimate layer, but when using the subset of feature-maps corresponding to $S$. Note that in all cases, the (one or more) feature-map activations are propagated to the penultimate layer using the pre-trained weights.

• From the variants of $\boldsymbol{C}$ we derive matching similarity matrices. The first, $\boldsymbol{Z}^{(\neg k)}$, is obtained by computing the Pearson correlation for each pair of rows in $\boldsymbol{C}^{(\neg k)}$. The second, $\boldsymbol{Z}^{(S)}$ is formed using the selected feature indices in $\boldsymbol{C}^{(S)}$.

• As indicated, $\boldsymbol{Z}_u$ and $\boldsymbol{H}_u$ denote the vectorized upper triangles of matrices $\boldsymbol{Z}$ and $\boldsymbol{H}$ respectively. The Pearson's correlation coefficient between the two is denoted as $\rho(\boldsymbol{Z}_u, \boldsymbol{H}_u)$. We adopt the terminology of referring to this value as a Baseline Second-Order-Isomorphism (2OI) between the two domains. Analogously, in some cases we compute $\rho(\boldsymbol{Z}_u^{(\neg k)}, \boldsymbol{H}_u)$ and $\rho(\boldsymbol{Z}_u^{(S)}, \boldsymbol{H}_u)$.

---

[1]Available at https://osf.io/kzxfg/

## 2.2 AIM 1: IDENTIFYING A SUBSET OF FEATURE MAPS THAT OPTIMIZES PREDICTION OF HUMAN SIMILARITY JUDGMENTS

We define the Alignment Importance Score (AIS) of each feature map in terms of its predictive capacity for the human representation $\boldsymbol{H}_u$. Intuitively, we aim to determine how the removal of each feature map $k \in \{1, 2, \ldots, 512\}$ affects the baseline isomorphism, $\rho(\boldsymbol{Z}_u, \boldsymbol{H}_u)$. The removal of each feature map produces a modified 2OI score, $\rho(\boldsymbol{Z}_u^{(\neg k)}, \boldsymbol{H}_u)$. Finally, The AIS of feature map $k$ is defined in Equation 1, with positive values indicating a relatively important feature map, and negative values a less important one. After computing AIS for all feature-maps, we rank-order them based on their AIS.

$$\text{AIS}_k = \rho(\boldsymbol{Z}_u, \boldsymbol{H}_u) - \rho(\boldsymbol{Z}_u^{(\neg k)}, \boldsymbol{H}_u) \tag{1}$$

We then identify an optimal subset of feature maps for predicting $\boldsymbol{H}_u$. In each iteration, one feature map is added to the subset $S$ in descending order of AIS rank, and we recompute the 2OI, $\rho(\boldsymbol{Z}_u^{(S)}, \boldsymbol{H}_u)$ using that subset. After these 512 iterations, subset $S^*$ ultimately selected is the one that maximizes 2OI.

To validate AIS, we use an 80:20 cross-validation framework where 80% of the entries in $\boldsymbol{H}_u$ are assigned to a training set, and the remaining 20% constitute the test set. The optimal subset of feature map indices, $S^*$, is determined from the training set using sequential features selection as described above. For statistical significance testing, we repeat the entire cross-validation process eight times with different dataset shuffling. This produces 40 Full vs. Retained value-pairs for each relevant comparison. To evaluate generalization, we use only this $S^*$ set of feature maps to predict HSJs on the test set. Prediction performance is compared against a baseline where all $d$ features are used for predicting HSJs in the test set. Statistical significance testing, per dataset, is based on the 40 value-pairs produced via cross-validation, which are analyzed using paired two-tailed T-tests (12 tests in all, non-corrected for multiple comparisons). Success of Aim 1 is determined if $\rho(\boldsymbol{Z}_u^{(S^*)}, \boldsymbol{H}_u)$ surpasses $\rho(\boldsymbol{Z}_u, \boldsymbol{H}_u)$, indicating superior prediction compared to the baseline using a subset of feature maps.

## 2.3 AIM 2: EXPLAINING HUMAN SIMILARITY JUDGMENTS

Our goal is to identify which image patches, in image space, are relevant to comparisons between a target image $t$ and other images in the set. This is visualized by creating a heatmap for $t$ identifying those image sections, as follows. We begin by defining a baseline 2OI for $t$ as the Pearson correlation between the $n - 1$ similarity judgments associated with $t$, as quantified from the model, and the corresponding set of human similarity judgments. As in Aim 1, we define the AIS of feature map $k$ by computing a value that reflects the departure from baseline, as indicated in Equation 1.

We iterate over all 512 feature maps, producing 512 AIS values that indicate the relative importance of each feature map for the alignment between DNN-derived distances and human similarity judgments for target image $t$. This produces, for each dataset containing 120 images, an $n \times k$ matrix (120 [AIS] x 512 [feature map]). We compare these distributions between the ImageNet and Ecoset-trained models to understand if and how the training regime impacts the distribution of AIS. Histograms are computed for the mean AIS value by feature, and the Mean Absolute Deviation, computed by feature (column) and by image (row).

Image-level heatmaps are then computed as follows. We first convert negative AIS values to zero because they indicate features that encode information less relevant to modeling the human data (see Eq. 1). The remaining scores are sum normalized. Subsequently, feature maps for an image are weighted-averaged according to their corresponding AIS to create a heatmap. In the heatmaps, warmer colors indicate image areas associated with the more important features.

To quantify the similarity between the heatmaps generated by Ecoset and Imagenet, we defined a Match score for each image as the Pearson correlation between the heatmap generated by the Ecoset model and the one generated by the Imagenet model. Anticipating the results, in certain instances, the Match score was low. We therefore examined if this occurred for images that did not correspond to classes on which the models were trained. For each image, we computed the entropy of the post-softmax probability distributions, independently for the Ecoset and Imagenet

trained models. The higher of these two entropy values was retained and designated as maxEntropy. Subsequently, considering all images in a dataset, we computed the correlation between the Match score and maxEntropy.

### 2.4 AIM 3: CROSS-REFERENCING HEATMAPS AGAINST SALIENCY MAPS

We compare the heatmaps produced by our method to those produced by TranSalNet Lou et al. (2022), which is a state-of-the-art DNN that identifies salient image sections and predicts well human gaze patterns. The following comparison was performed for each image. First, for each image we create a heatmap as described in Aim 2, and a saliency map from TranSalNet. We keep the same aspect ratio of the images input to both VGG-16 and TranSalNet for compatibility in later comparisons. We aim to identify whether an image section (specifically, a pixel) identified as salient (*Sal*) is more likely to also be identified as comparison-relevant (*CR*; that is, warm-colored in our analysis). To do this we threshold both maps to select the top 5% of Salient and *CR* pixels, producing $Sal$, $\neg Sal$, $CR$ and $\neg CR$ partitions of the image pixels. We then compute the Relative Risk (RR) ratio as in Equation 2.

$$\text{RR} = P(CR|Sal) \div P(CR|\neg Sal) \tag{2}$$

The relative risk as computed here measures the likelihood of *Sal* pixels being *CR* pixels compared to $\neg Sal$ pixels. We repeat this analyses when thresholding both maps at 10% and 15% top *Sal* and *CR* pixels.

We note that there is no requirement that the two methods identify the same image features. The saliency map is driven by image features (including higher level semantics captured by the DNNs), whereas the heatmap we produce from AIS values is a function of how a certain object stands in relation to other objects in the set. As we will see, this produces cases of very high overlap, but also important distinctions.

## 3 RESULTS

### 3.1 AIM 1: IDENTIFYING A SUBSET OF FEATURE MAPS THAT OPTIMIZES PREDICTION OF HUMAN SIMILARITY JUDGMENTS

As shown in Figure 1, by computing AIS it was possible to identify a subset of 512 feature maps for each dataset, which produced improved out-of-sample predictions compared to a baseline condition where all feature maps were used. This was consistent for models trained on Ecoset or ImageNet, with less than 50% of the 512 feature maps being used in two cases. Paired T-tests indicated that of the 12 comparisons between predictions from Full and Retained features, ten were statistically significant (p-values $< 0.01$). The two cases where pruning did not produce improvement were the Transportation and Vegetables datasets for the ImageNet-trained model. A divergence in performance between ImageNet and Ecoset was found for Animals and Transportation, where ImageNet produced markedly better predictions. This is attributable to the distribution of such images in the training set and the definition of classes used for training.

Speaking to category-specific information, AIS values for each feature-map differed across datasets. That is, feature maps important for aligning one category were not necessarily important for another category. To evaluate this issue, we computed pair-wise Pearson correlations between the AIS values of the 512 feature-maps for each pair of datasets (e.g., Fruits vs. Vegetables). For both Ecoset and ImageNet, the strongest correlation was between Fruits and Vegetables (ImageNet $R = 0.29$; Ecoset $R = 0.26$). For Ecoset, the second highest correlation was between Transportation and Furniture ($R = 0.23$), whereas for ImageNet it was between Various and Animals ($R = 0.16$). All other correlations, in both analyses, were below 0.1.

### 3.2 AIM 2: EXPLAINING HUMAN SIMILARITY JUDGMENTS

Figure 2 shows examples of heatmaps produced by alignment importance scoring. Given that each dataset contained 120 images, we selected 4 images from each dataset according to the principle that two of the images produced apparently sensible results, and the two others were less sensible.

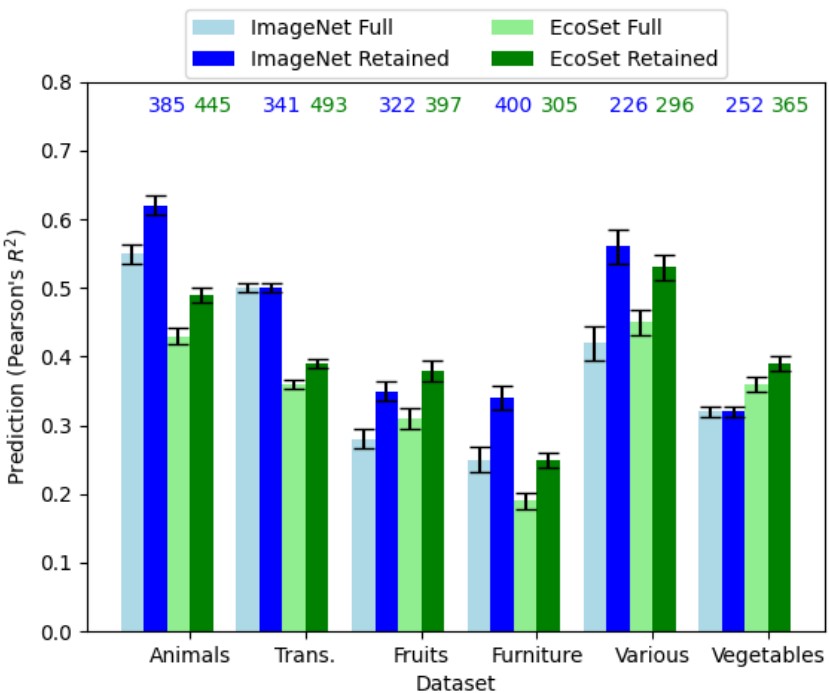

Figure 1: Out-of-sample predictions of human similarity judgments using image embeddings. Full: using all 512 feature maps. Retained: using feature maps identified from an independent training set. The numbers above the second and fourth columns in each group represent averages of feature-map set sizes across 40 folds. Error bars indicate standard errors adjusted for paired-comparisons (Loftus & Masson, 1994).

Figure 3 shows, for each dataset, histograms computing the Average AIS associated with each feature (log10 scaled), and the Mean Absolute Deviation computed per feature (column) and per image (row). The histogram shows that the average AIS rarely exceeded 0.01 for any feature (Figure 3a). Two-sided Kolmogorov-Smirnov tests (Hodges Jr, 1958) were conducted to verify if the histograms associated with the two training regimes (ImageNet, Ecoset) came from the same distribution. Overall, the test shows that the profiles were similar in the case of average values (all p-values $\geq 0.05$), except for the Furniture dataset, $p = .002$. With respect to Mean Absolute Deviation (MAD), when computed per feature (Figure 3b) we find that the values varied around one order of magnitude, with a few features showing relatively higher values meaning they were much more important for some images than others. The MAD histograms computed from per-image data indicated that ImageNet's AIS distribution was consistently left shifted with respect to Ecoset's (Figure 3c). This means that the AIS produced by Ecoset-trained model are more spread out, suggesting a more meaningful separation between those features relevant for alignment and those that are not. Two-sided Kolmogorov-Smirnov tests on Mean Absolute Deviation verify significant differences between the two models in all cases (all six datasets, KS tests, $p < .05$).

To assess the similarity of heatmaps produced by Ecoset and ImageNet, for each image we calculated the correlation between the heatmaps produced by the two methods. The median correlation values were as follows: $0.82 \pm 0.17$ for Animals, $0.72 \pm 0.17$ for Transportation, $0.73 \pm 0.22$ for Fruits, $0.68 \pm 0.24$ for Furniture, $0.66 \pm 0.23$ for Various, and $0.64 \pm 0.25$ for Vegetables. In all datasets the maximum correlation values approached 1.0, while the minimum values often approached zero. This means that although agreement was often good, training models on Ecoset or ImageNet can produce different heatmaps. These findings are consistent with those of Aim 1, which showed that the VGG-16 models trained on the two datasets capture and learn human similarity judgments in slightly different ways.

As detailed section 2.3, we evaluated if images that presented a lower Match between Ecoset and ImageNet heatmaps were associated with higher entropy of post-softmax values in either of the

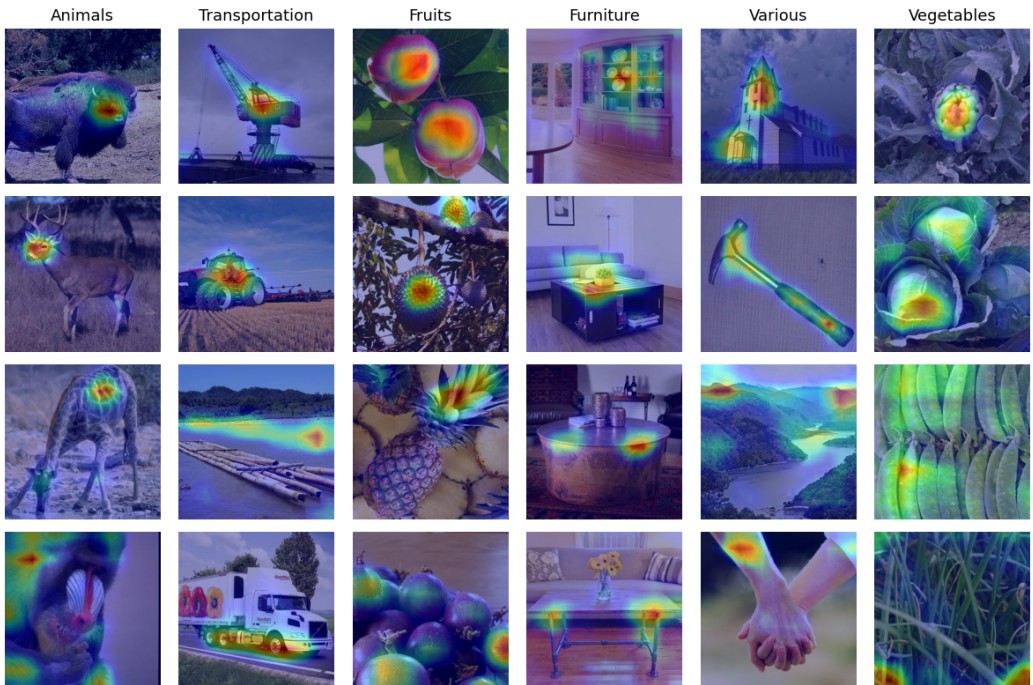

Figure 2: Heatmaps generated using Alignment Importance Scores of feature maps trained with Ecoset. For each dataset, two images with subjectively higher interpretability (top two rows) and lower interpretability (bottom two rows) were selected.

two sets (maxEntropy), which would produce a negative correlation between the two quantities. We found that this was indeed the case, for Animals ($R = -0.39$), Fruits ($R = -0.37$), Various ($R = -0.29$), and Vegetables ($R = -0.31$). Thus, images that do not present information sufficient for classification produce disagreement between the two models. These might be out of distribution images or bad examples of trained categories. Ultimately, in those cases where heatmaps differ, the results of Aim 1 may be used as a guide to inform whether Ecoset or ImageNet is more plausible with respect to the human representation of a given category.

### 3.3 AIM 3: CROSS-REFERENCING HEATMAPS AGAINST SALIENCY MAPS

Table 1: Relative Risk values comparing heatmaps computed from Alignment Importance Scores to those generated by TranSalNet, a saliency model that predicts human gaze. Chance values are $RR = 1$.

| Category | Ecoset | | | ImageNet | | |
|---|---|---|---|---|---|---|
| | 5% vs. 5% | 10% vs. 10% | 15% vs. 15% | 5% vs. 5% | 10% vs. 10% | 15% vs. 15% |
| Animals | $31.9 \pm 34.5$ | $17.2 \pm 18.1$ | $12.2 \pm 10.3$ | $33.6 \pm 48.1$ | $16.4 \pm 14.5$ | $11.9 \pm 9.0$ |
| Transportation | $7.4 \pm 11.5$ | $6.9 \pm 10.4$ | $6.0 \pm 8.3$ | $5.0 \pm 7.5$ | $4.7 \pm 5.6$ | $4.9 \pm 5.4$ |
| Fruits | $9.2 \pm 16.5$ | $7.9 \pm 13.5$ | $6.2 \pm 8.2$ | $9.8 \pm 20.5$ | $7.1 \pm 11.5$ | $6.0 \pm 9.0$ |
| Furniture | $7.5 \pm 13.1$ | $5.7 \pm 7.2$ | $4.8 \pm 5.1$ | $7.0 \pm 12.0$ | $5.6 \pm 7.4$ | $4.8 \pm 4.6$ |
| Various | $15.2 \pm 22.6$ | $9.6 \pm 10.3$ | $8.0 \pm 7.5$ | $16.5 \pm 33.9$ | $9.4 \pm 10.6$ | $7.8 \pm 8.2$ |
| Vegetables | $7.6 \pm 12.5$ | $5.6 \pm 7.6$ | $4.6 \pm 4.8$ | $7.0 \pm 13.9$ | $4.9 \pm 6.4$ | $4.0 \pm 4.1$ |
| All datasets | $13.1 \pm 22.0$ | $8.8 \pm 12.4$ | $7.0 \pm 8.0$ | $13.2 \pm 28.5$ | $8.0 \pm 10.7$ | $6.6 \pm 7.5$ |

We generally observe good agreement between our heatmaps and the saliency maps produced by TranSalNet, as indicated by the Relative Risk values strongly exceeding 1.0 (Table 1). This means that pixels identified as highly salient by TranSalNet were more likely to be associated with comparison-relevant (warm colors) pixels in our heatmaps than were pixels identified as not highly

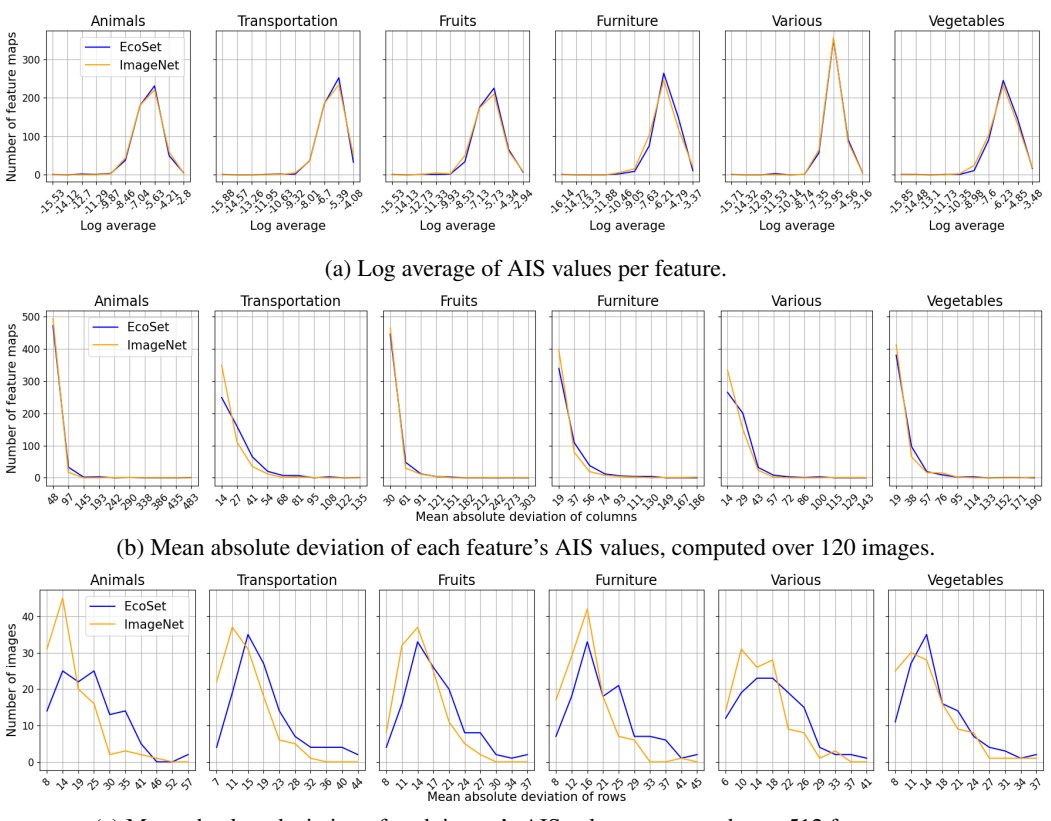

(a) Log average of AIS values per feature.

(b) Mean absolute deviation of each feature's AIS values, computed over 120 images.

(c) Mean absolute deviation of each image's AIS values, computed over 512 feature maps.

Figure 3: Histograms describing statistics of Alignment Importance Score distributions for models trained on Ecoset or ImageNet. The x-axis of (b) and (c) are displayed in e-4 format.

salient. This was found regardless of whether pixels in both heatmaps were thresholded at top 5%, top 10% or top 15%. As the table shows, the values often exceeded 10, reaching as high as 30 for Animals. The data were quite similar for ImageNet and Ecoset overall, with the exception of Transportation, for which Ecoset appeared to produce heatmaps more strongly associated with saliency maps. Furthermore, RR varied significantly across categories, being highest for Animals, and lowest for Vegetables. This suggests that for Animals, elements salient in images are also important for comparison, whereas this is less so for Vegetables.

Figure 4 presents images on which we plotted contours reflecting TranSalNet's salience (orange) and alignment score heatmaps (blue) to visualize their overlap. For the two images on the left (bison and crane), the salience and alignment maps consistently show strong agreement across all three thresholding levels. For the two right images, there is no overlap. Specifically, the monkey's facial features are highly salient, but are not identified as important for alignment. In the case of the truck image, the banner area depicting colorful peppers is identified as salient, but the wheel area is identified as important for alignment. This is reasonable, as means of transportation in the set are effectively compared by observing the lower section of the vehicle, which differentiates trucks, cars, buses, motorcycles, trains and so on. Indeed we find these elements are often highly salient in the produced heatmaps. More results with appropriate level of detail are shown on Figshare[2] or in the Appendix section below.

## 4 Conclusion

Understanding what information is used in human comparisons is important not only for a better understanding of the comparison process itself, but also for comprehending how people form memories

---

[2]https://figshare.com/s/506d26037e071612cdf3

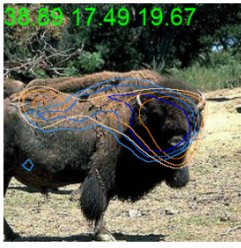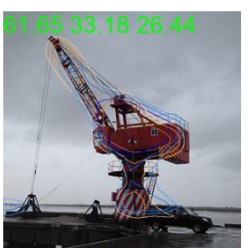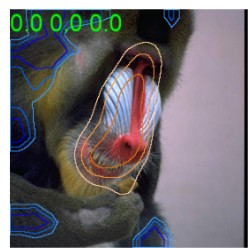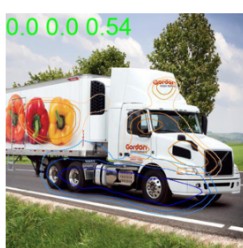

Figure 4: Overlap between the heatmaps created by Alignment Importance Scores (blue contours) and the saliency maps from TranSalNet (orange contours). The contours indicate the 5%, 10%, and 15% most important pixels, with increasing color intensity respectively. Relative Risk values computed from top 5%, 10% and 15% pixels in each map are printed on the top of each images. The two left images are examples of cases where AIS and saliency identified similar areas, whereas the two right images present extreme cases of non-overlap.

and make decisions (Roads & Love, 2024). We introduced and validated a feature-map's Alignment Importance as a meaningful parameter by showing that it generalizes to improve prediction of human similarity judgments. This complements current approaches that achieve improvements by using reweighting or pruning of nodes in a DNN's penultimate layer (e.g., Peterson et al., 2018; Attarian et al., 2020; Kaniuth & Hebart, 2022; Jha et al., 2023; Tarigopula et al., 2023; Flechas Manrique et al., 2023). We then used AIS to produce explanations for those judgments via heatmaps, which often corresponded quite well to state-of-the-art saliency maps. However, instances where the two maps diverge show it is possible to dissociate visually salient image elements from those that are important for comparison. Since the current selection of feature maps is greedy, future works may incorporate heuristic information into the procedure to avoid this problem. Furthermore, quantifying the correlation among feature maps prior to computing AIS could potentially improve the assessment of their cognitive relevance compared to human data.

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

# A    APPENDIX

We present more images to show the overlap between the heatmaps created by Alignment Importance Scores (blue contours) and the saliency maps from TranSalNet (orange contours). As described in the result section above, the contours indicate the 5%, 10%, and 15% most important pixels, with increasing color intensity respectively. Relative Risk values computed from top 5%, 10% and 15% pixels in each map are printed on the top of each images.

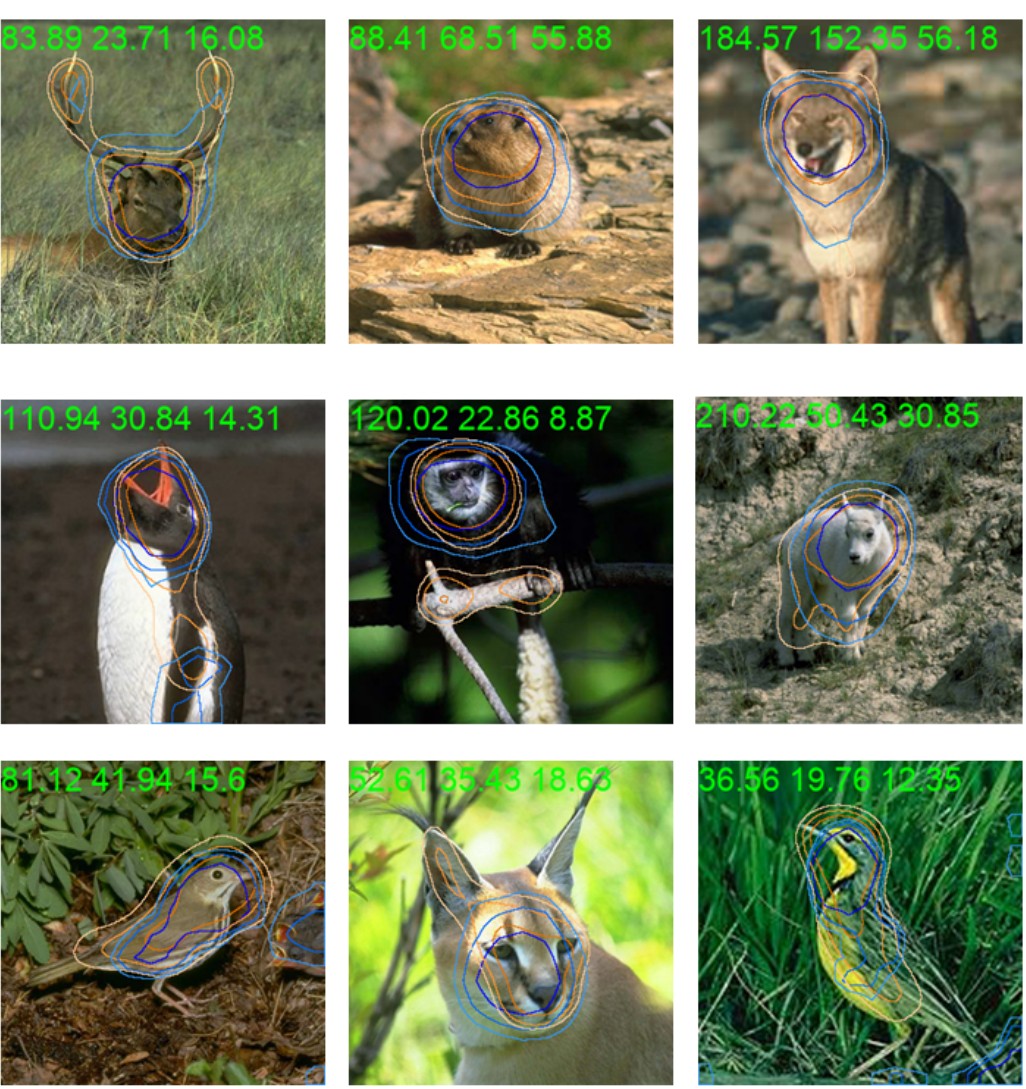

# Animals, Low Relative Risk Ratio

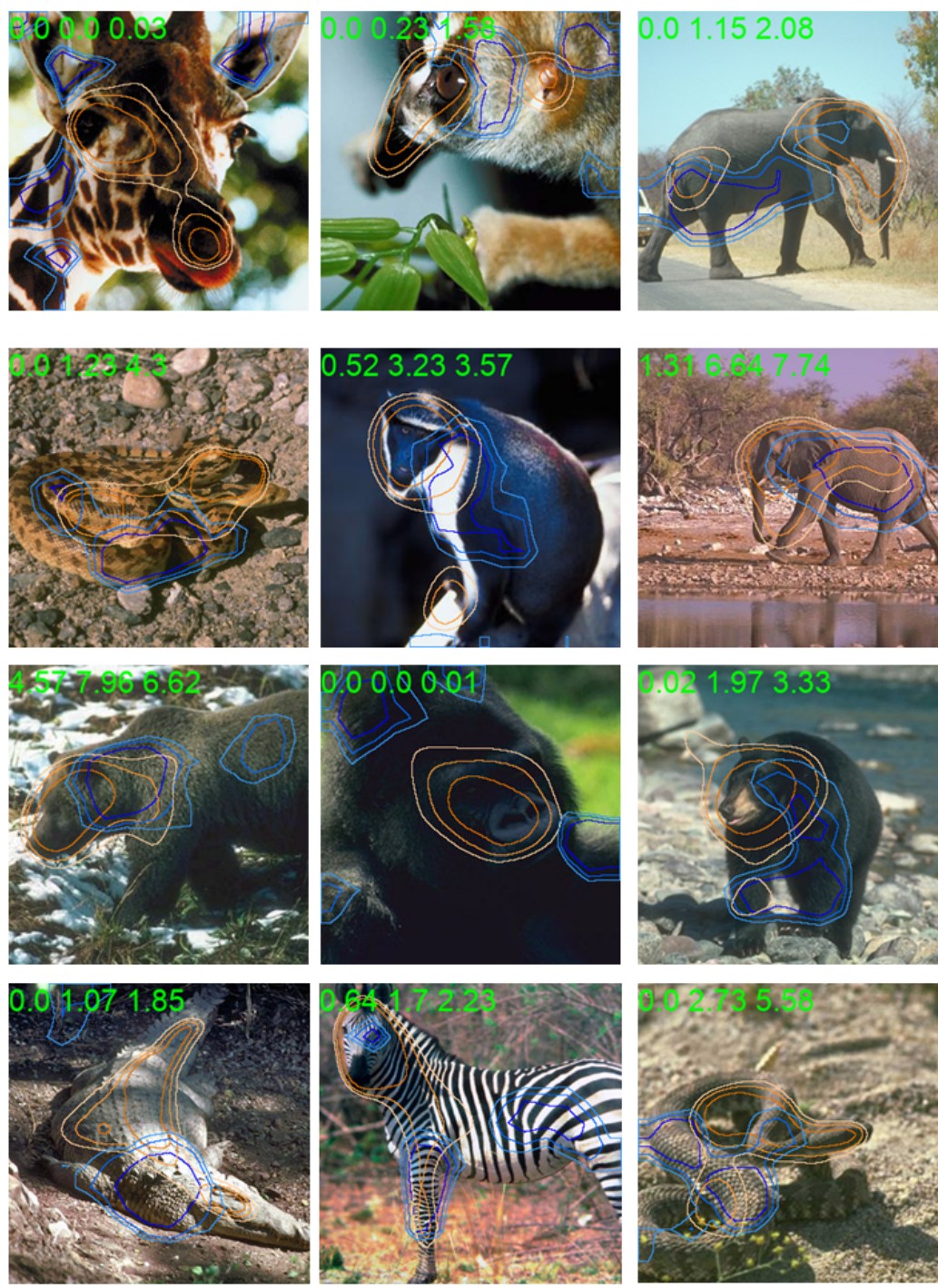

## **Furniture**, High Relative Risk Ratio

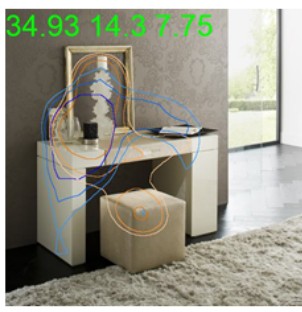
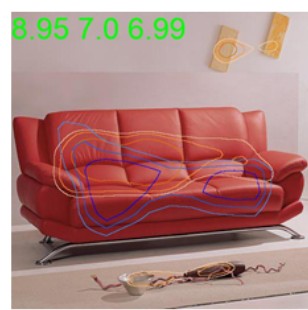
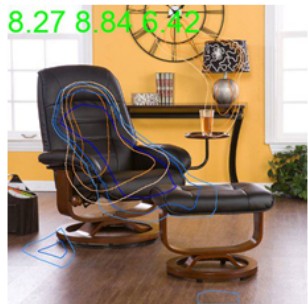

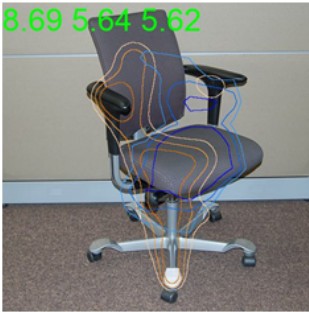
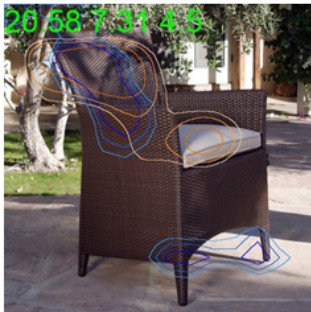
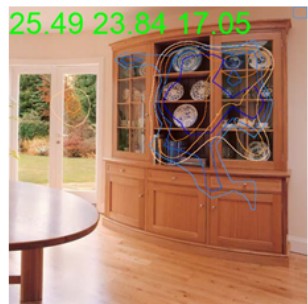

## **Furniture**, Low Relative Risk Ratio

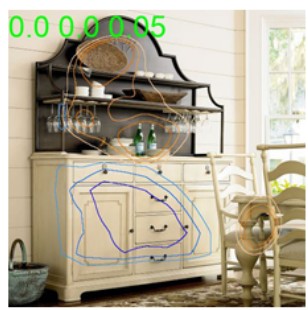
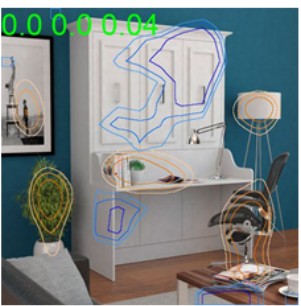
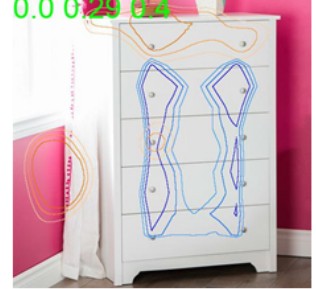

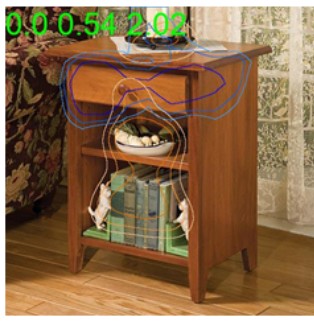
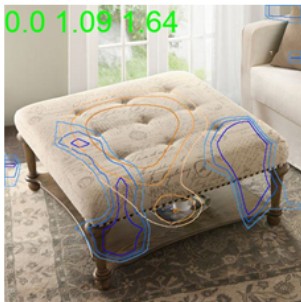
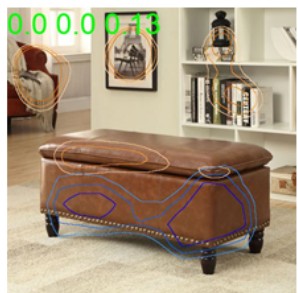

