# OpenReview forum: "Explaining Human Comparisons using Alignment-Importance Heatmaps"
_ICLR.cc/2024/Workshop/Re-Align — ICLR 2024 Workshop Re-Align Poster_

### Official Review · Reviewer_t7jh · 2024-02-24
**Interesting motivation and problem; needs improvements in clarity/presentation**

**Rating:** 1
**Fit:** 3
**Confidence:** 1

**Workshop Review:**

From what I understand, the Alignment Importance Score can be a useful metric for measuring network features' predictive capacity of human representations. The paper presents interesting approaches to better predictions and understanding of human similarity judgments. While I can see the novelty and interest to the community in the presented motivation and problem space, I think the clarity and presentation needs to be improved before publication. The sections (especially "Logic of Approach") read a bit as stream-of-consciousness writing and makes it difficult for me to fully appreciate the contribution. I apologize for the lack of specificity in this review regarding content and experimental design; while I would love to give more feedback in these areas, I had a hard time following the train of thought in the writing.

**Reason For Not Giving Higher Score:**

Needs a re-organization of writing and figures for clarity of method (see above review for details about improvements).

**Reason For Not Giving Lower Score:**

N/A

**Reviewer Domain:**

machine learning

---

> ### Author Response · Authors · 2024-05-03
>
> Thank you for taking the time and effort to review our manuscript. We have made modifications to that section by splitting it into 2 subsections, and providing additional clarifications on the logic of the study (subsection 1.2), as well as elaborating on the aims and contributions (subsection 1.3).

---

### Official Review · Reviewer_XFUg · 2024-02-25
**masking for explanation of alignment**

**Rating:** 3
**Fit:** 3
**Confidence:** 2

**Workshop Review:**

Nice work. How masking is used is fairly clear, though I wonder if it hits limits when  non-contiguous regions are key to the alignment only in the context of each other. E.g., if one is masked, the other can play the same role.

Ideally, there would be more comparison to other techniques.

Potential advantage of explaining what is driving the judgment.

**Reason For Not Giving Higher Score:**

NA

**Reason For Not Giving Lower Score:**

I have doubts on how well masking of regions in isolation will do in providing clear interpretations, but the work is interesting and relevant.

**Reviewer Domain:**

cognitive science

---

> ### Author Response · Authors · 2024-05-03
>
> Thank you for taking your time and effort to give constructive feedback. The reviewer is raising two related points:
>
> 1. What if non-contiguous regions are necessary. We clarify that because we use an entire feature map as the explanation unit, non-contiguous image regions are naturally captured in each feature map, that is, each feature map in principle spans the entire image area.
>
> 2. What happens in the case that two feature maps are correlated and so if one is masked, the other plays the same role, which, on our current computation, would result in a low AIS score as removal of one feature map will have no impact. This problem can be solved, in principle, using forward feature selection (rather than backward selection which we implemented), where the importance of each feature is computed when that feature alone is used for prediction of the target variable. The practical problem in implementing this method in the current study is that in this case, each feature produces very small $R^2$ values, which appears to be a weak point for further computations. Another possibility is to use nested backward selection where the feature ranking is recomputed after removal of each feature. However, this method would require significantly more time to complete each pass, as it requires exponential rather than linear time.

---

### Decision · Program_Chairs · 2024-03-02

Accept (Poster)